# Multi-class classification algorithms for the diagnosis of anemia in an outpatient clinical setting

Rajan Vohra[1]*, Abir Hussain[2,3], Anil Kumar Dudyala[4], Jankisharan Pahareeya[5], Wasiq Khan[3]

1 School of Creative Technologies, University of Bolton, Bolton, United Kingdom, 2 Department of Electrical Engineering, University of Sharjah, Sharjah, UAE, 3 Department of Computer Science, Liverpool John Moore's University, Liverpool, United Kingdom, 4 Department of Computer Science, National Institute of Technology Patna (NIT Patna), Patna, India, 5 Department of Information Technology, Rustamji Institute of Technology, BSF Academy, Tekanpur, Gwalior, India

* rv1res@bolton.ac.uk

**Data Availability Statement:** Data is held in the Mendeley Data repository. DOI: 10.17632/dy9mfjchm7.1 URL: https://data.mendeley.com/datasets/dy9mfjchm7/1.

## Abstract

Anemia is one of the most pressing public health issues in the world with iron deficiency a major public health issue worldwide. The highest prevalence of anemia is in developing countries. The complete blood count is a blood test used to diagnose the prevalence of anemia. While earlier studies have framed the problem of diagnosis as a binary classification problem, this paper frames it as a multi class (three classes) classification problem with mild, moderate and severe classes. The three classes for the anemia classification (mild, moderate, severe) are so chosen as the world health organization (WHO) guidelines formalize this categorization based on the Haemoglobin (HGB) values of the chosen sample of patients in the Complete Blood Count (CBC) patient data set. Complete blood count test data was collected in an outpatient clinical setting in India. We used Feature selection with Majority voting to identify the key attributes in the input patient data set. In addition, since the original data set was imbalanced we used Synthetic Minority Oversampling Technique (SMOTE) to balance the data set. Four data sets including the original data set were used to perform the data experiments. Six standard machine learning algorithms were utilised to test our four data sets, performing multi class classification. Benchmarking these algorithms was performed and tabulated using both 10 fold cross validation and hold out methods. The experimental results indicated that multilayer perceptron network was predominantly giving good recall values across mild and moderate class which are early and middle stages of the disease. With a good prediction model at early stages, medical intervention can provide preventive measure from further deterioration into severe stage or recommend the use of supplements to overcome this problem.

## Introduction

Anemia is a disease caused by the deficiency of Iron and it is one of the most critical health problems globally causing serious public health issue [1]. According to the World Health

**Funding:** The authors received no specific funding for this work.

**Competing interests:** The authors have declared that no competing interests exist.

Organization (WHO), Anemia prevalence of over 40% in a community makes it a public health issue of critical importance [2]. Anemia prevalence in children may be due to genetic reasons or due to deficiencies in nutrition such as iron, folate, vitamins A/B12 and copper. Iron deficiency is the most important determinant of Anemia [3]. Other factors include socio demographic characteristics of mother households such as region, wealth index, water sources, working status, while Anemia status along with child features like age, nutritional status, and weight at birth are the most critical features influencing Anemia in the children age group of 6–59 months [4]. According to the WHO, Anemia prevalence occurs in most of the countries in Africa and South Asia and some countries in East Asia and the Pacific. While the highest prevalence of Anemia is found in Africa, the largest numbers of children affected by Anemia are found in Asia [5]. Machine learning is the process of discovering novel patterns or associations while analyzing large data sets [6]. Machine learning is being increasingly used in analysis and prediction of diseases. In this framework, we find that classification techniques and algorithms are the most potent of machine learning algorithms that are contributing immensely to this endeavor [7, 8]. In healthcare and medical research, many machine learning algorithms are used for the prediction of diseases in study populations.

Our extensive researches indicated that machine learning techniques such as support vector machines (SVM), Random Forest and Artificial Neural Networks (ANN) have been widely applied for the classification of various diseases such as Diabetes [9–11], Appendicitis [12], and Multiple Sclerosis (MS) [13]. Machine learning techniques to classify Anemia in children are still evolving. Along with traditional clinical practices, machine learning techniques can be utilized to predict the risk of Anemia prevalence in children. Some key research in this direction has been undertaken as demonstrated in reference [14, 15], which have constructed prediction models for Anemia status in children.

The prevalence of Anemia among adults was studied in [16] by taking complete blood count (CBC) at a referral hospital in Southern Ethiopia. Prevalence and severity were related with age and gender and were analyzed. Chi square test was used to analyze the statistical significance for association among categorical variables [16]. In fact, this research paper extends this work by taking the collected data set as a base and then applies Multi class classification algorithms to identify the class label of the diagnosed patients as–Mild, Moderate and Severe. Public health is influenced by many factors such as socio economic, demographic, behavioral and other factors related to the health care system [17]. Social factors such as income, wealth, education can affect health markers in people such as blood pressure, body mass index (BMI), and waist size etc. [18]. In a study by Luenam et al [19], multiple logistic regressions is used to analyze the relationship/ causality between respiratory diseases that are chronic and socio economic factors like gender, region, cooking fuel used and smoking / nonsmoking status [19]. Multiple approaches in data mining have been used to predict malaria and anemia among children using diverse data sets [20, 21]. According to a study done by Rosangela et al [22], it is shown using regression analysis that factors like age, nutrition, income, parental educational status, the child's environment and previous illnesses are the social determinants of iron deficiency anemia [22]. Sow et al [23] used support vector machines (SVM) and demographic health survey data from Senegal to classify malaria and anemia status accurately. Sow et al studied anemia and malaria classification problems using demographic and health survey data from Senegal. Using feature selection, the number of features of both anemia and malaria data sets were reduced. There after using Variable Importance in Projection (VIP) scores, the relative importance of social determinants for both anemia and malaria prevalence were computed. Finally using machine learning algorithms the classification for both anemia and malaria were undertaken–Artificial neural networks (ANN), KNN (K nearest neighbors), Random Forests, Naïve Bayes and Support vector machines (SVM) were used [24]. Lisboa has

demonstrated the utility and potential of Artificial neural networks (ANN) in health care interventions [25]. Children under 5 years old and pregnant women are more vulnerable to anemia due to the greater requirements of Iron for body growth and the expansion of Red Blood Cell (RBC) [26]. Iron deficiency anemia results in a reduction in academic performance and work capacity, which reduces the earning potential of individuals, which affects national economic growth [27]. Alemayehu has used multi variable logistic regression to study the magnitude and severity of anemia in a predominantly rural setting in Ethiopia. Mothers age, mother's occupation, gender of child, Dietary Diversity and food security were found to be key factors influencing the prevalence of anemia in the community population studied [28]. Using complete blood count (CBC) samples, a study to classify anemia using machine learning algorithms of Random Forests, C4.5 (Decision tree), and Naïve Bayes (NB) was undertaken. Comparison of the classifier algorithms for mean absolute error (MAE) and classifier accuracy were computed and tabulated [29]. Laengsri et al [30] used KNN, Decision tree (J48 algorithm), Artificial Neural Networks (ANN) and Support Vector Machines (SVM) to classify computations to distinguish iron deficiency anemia and Thalassemia in a study conducted in Thailand. This also resulted in the development of a web based tool (ThalPred) for this computation [30]. Understanding the geographical distribution of anemia can help target prevention and control mechanisms in which spatial distribution is mapped along with determinant factors in a study done in Ethiopia [31]. The spatial pattern of the rate of anemia among women of reproductive age was visualized and a spatially smoothed proportion obtained using empirical Bayes estimation methods for the spatial analysis [32].

There are three key computations that define the novelty of this research paper.

This paper uses complete blood count (CBC) data to diagnose the level of Anemia prevalent in the study population. Earlier studies have framed the diagnosis task as a binary classification problem–the patients in the study population are either anaemic or non-anaemic.

In this paper the diagnosis is framed as a multi class classification problem. We use feature selection to identify key attributes of the patient data set. We have used Majority voting on three feature selection techniques–correlation [33], Classification and Regression Tree (CART) [34] and Gradient Boosting [35] to identify the key attributes of the input patient data set.

We benchmark the performance of key machine learning algorithms for multi class classification. The computations are done using both the 10-fold cross validation method as well as the Hold out method. The results of this analysis are tabulated as described further in this paper.

The rest of this paper is organized as follows. Section 2 material and methods, Section 3 data analysis while Section 4 describes feature selection method used in this paper. Section 5 presents work flow, Section 6 describes techniques employed for the anemia classification. Section 7 presents performance assessment parameters used, while Section 8 presents the results and discussions. Finally, Section 9 presents conclusions and directions for further work.

## Material and methods

We assess the prevalence of different types of Anemia including its severity and association with age and gender of the study population with CBC data set parameters as variables. We use data from complete blood count test performed by Hematology analyzer to determine the prevalence of different types of anemia treated at the Eureka diagnostic center in Lucknow, India. All the procedures for the CBC test were done following standard operating protocols defined for the Hematology analyzer.

## Study subjects

The study subjects were patients who visited the Eureka diagnostic center, Lucknow, India for CBC investigation, of whom 400 patient samples were randomly selected to compute the prevalence of anemia and for further investigation into Anemia classification. The patients visited the Eureka diagnostic center in Lucknow for various clinical examinations. The diagnostic center performs 4 – 8CBC investigations a day on average. During the data collection period, between September 2020 to December 2020, 1000 CBC investigations were performed, out of which 400 random samples were selected for further analysis and study. Data set was available on **Mendeley Data** Repository, Data identification number: 10.17632/dy9mfjchm7.1

Direct URL to data: https://data.mendeley.com/datasets/dy9mfjchm7/1.

## Inclusion criterion

Hemoglobin values represented by the HGB attribute in the CBC data set was selected as the response variable. We included adult males and females who are not pregnant and older than 15 years of age in the study population. Infants and young children less than 10 years old and pregnant women were excluded from the study due to various factors like variable CBC test values and other factors. After excluding the above stated persons from the randomly chosen sample of 400 patients, we were left with 364 patients in the final data set to be investigated.

## Procedures

Laboratory staff of Eureka diagnostic center collected 5 ml of blood sample for the CBC tests. Randomly selected samples of patients for this study had HGB values between 4.2and 19.6and they were selected for further analysis.

The CBC report had 11 attributes in the data set shown in the Table 1 Titled Dataset Description.

These attribute values were recorded for each patient in the sample data set. The computation starts with an analysis of the distribution of anemia as describes in the S1 Appendix.

## Ethical considerations

Ethical consideration was approved by Eureka diagnostic center, Lucknow, India to collect and analyze patient data for the CBC test reports in which patient consent was obtained. Support for the data collection and compilation was also obtained from the management and staff

**Table 1. Dataset description.**

| SN | Attributes | Attribute Characteristics | Abbreviation |
|---|---|---|---|
| 1 | Age | numerical | age |
| 2 | Gender | character | gender |
| 3 | Hemoglobin | numerical | HGB |
| 4 | Mean cell volume | numerical | MCV |
| 5 | Mean cell hemoglobin | numerical | MCH |
| 6 | Mean cell hemoglobin concentration | numerical | MCHC |
| 7 | Red cell distribution width | numerical | RDW |
| 8 | Red blood cell count | numerical | RBC |
| 9 | White blood cell count | numerical | WBC |
| 10 | Platelet count | numerical | PLT |
| 11 | Packed cell volume | numerical | PCV |

of Eureka diagnostic center during the data collection which took place in the period September 2020 to December 2020.

## Data analysis

In the eleven attributes of the Complete Blood Count (CBC) patient data set the Hemoglobin (HGB) attribute is the response variable. The anemia status in an individual patient can be either mild, moderate or severe based on the value of the HGB variable. This categorization for both men and women is described in S1 Appendix. This is defined strictly according to the World Health Organization (WHO) guidelines for anemia categorization.

Data collected was entered in Excel format for the eleven attributes with the size of the data set being 364 records. The prominent feature selection techniques such as Correlation, CART and Recursive feature selection techniques using gradient boosting were used to identify significant features (feature selection). The brief description of the above techniques can be seen in the section named Feature Selection. We performed majority voting for feature selection. Weka tool was used to compute the multiclass classification output and the final results were computed and tabulated [36]. Our dataset was imbalanced in which imbalance class creates a bias where the machine learning model tends to predict the majority class [37]. We used Synthetic Minority Oversampling Technique (SMOTE) techniques for balancing the data set using Knime tool available at https://www.knime.com/downloads.

Association of anemia prevalence with age and gender were computed. The prevalence of microcytic, normocytic and macrocytic anemia was computed along with their association with Age. This is shown in Tables 2 and 3 along with Figs 1 and 2.

In addition Feature selection was performed using multiple techniques to identify the important features of the dataset which was then further studied using multiclass classification to determine the extent of anemia prevalence as being Mild, Moderate or Severe.

The final results obtained were compiled and prepared in the form of this research paper to report the findings of this study.

A total of 364 patient samples constituted the study population for this research study. We note that mild anemia is most prevalent in the 46–60 years age group, moderate anemia is prevalent most in the younger age group of 10–30 years age group, while severe anemia is most prevalent in the 61–90 years age group.

It was noted that from Figs 1 and 2, mild anemia occurs lowest in the 61–90 years age group. Moderate anemia occurs lowest in the 31–45 years age group. Severe anemia has lowest prevalence in the 46–60 years age group. The results of the data description presented in Table 2 read in conjunction with Fig 3 describe the distribution of normocytic, microcytic and macrocytic anemia.

It can be observed that Normocytic anemia is highest in prevalence in the 10–30 years age group and lowest in the 61–90 years age group. Microcytic anemia is highest in prevalence in the 61–90 years age group and lowest in the 10–30 years age group. Macrocytic anemia is

**Table 2. Prevalence of mild, moderate and severe anemia by age.**

| Age (years) | percentage (%) of anemia | | |
|---|---|---|---|
| | mild | Moderate | severe |
| 10–30 | 66.7 | 30.5 | 2.8 |
| 31–45 | 75 | 20 | 5 |
| 46–60 | 75.2 | 22.5 | 2.3 |
| 61–90 | 65 | 26.6 | 8.4 |

**Table 3. Prevalence of normocytic, microcytic and macrocytic by age.**

| age (years) | percentage (%) of anemia | | |
|---|---|---|---|
| | normocytic | microcytic | macrocytic |
| 10–30 | 75 | 17.59 | 7.407407407 |
| 31–45 | 16.25 | 75 | 8.75 |
| 46–60 | 13.9784946 | 80.64516129 | 5.376344086 |
| 61–90 | 13.253012 | 81.92771084 | 4.819277108 |

highest in prevalence in the 31–45 years age group and lowest in the 61–90 years age group. We further note that the 10–30 years age group has most common occurrence of normocytic anemia, the 31–45 years age group has microcytic as the most common prevalence, the 46–60 years age group has microcytic as the most common prevalence and the 61–90 years age group has microcytic as the most common prevalent anemia. Furthermore, the second most prevalent anemia for the age group 10–30 years was microcytic, 31–45 years was normocytic, 46–60 years was normocytic and 61–90 years was normocytic. Table 3 shows the prevalence of normocytic, normochromic and macrocytic by age while Table 4 gender wise prevalence of anemia in the study population. The overall prevalence of anemia in the given data set is as follows:

Mild anemia: 70.32%

Moderate anemia: 25.28%

Severe anemia: 4.40%

This shows that mild anemia is most prevalent in the study population. In females, the most prevalent is the mild anemia and in males the most prevalent is also the mild anemia. Moderate anemia is more prevalent in males than in females. Severe anemia is more prevalent in males than in females.

## Feature selection

The feature selection methods are utilised to reduce redundant features extracted from the raw data. The objective is to provide better understanding of the dataset and to allow a faster

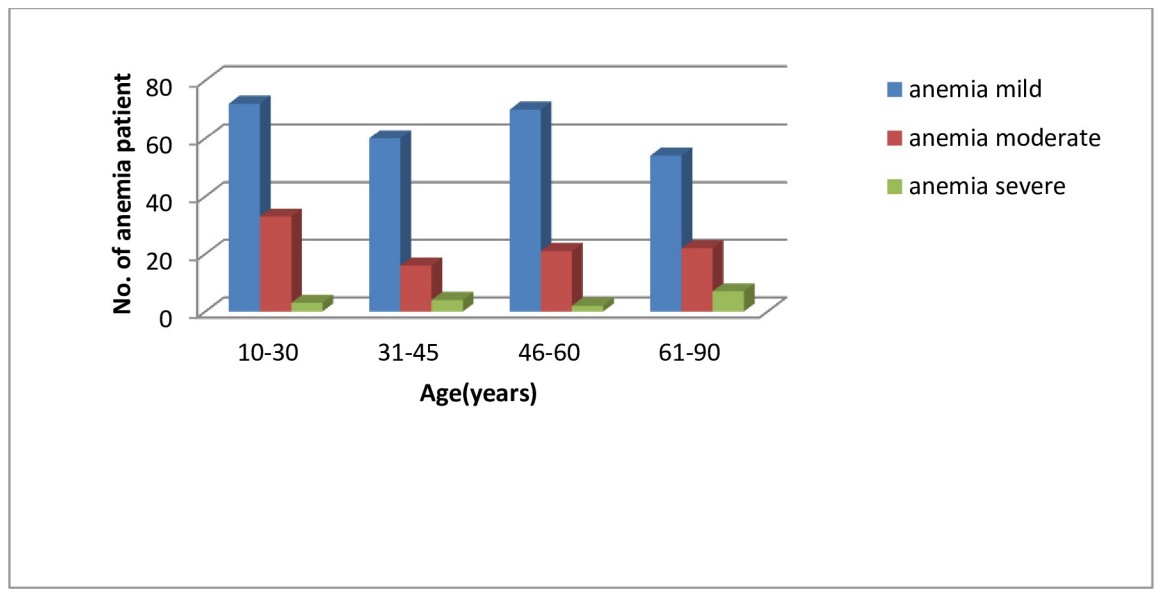

**Fig 1. Prevalence of mild, moderate and severe anemia by age.**

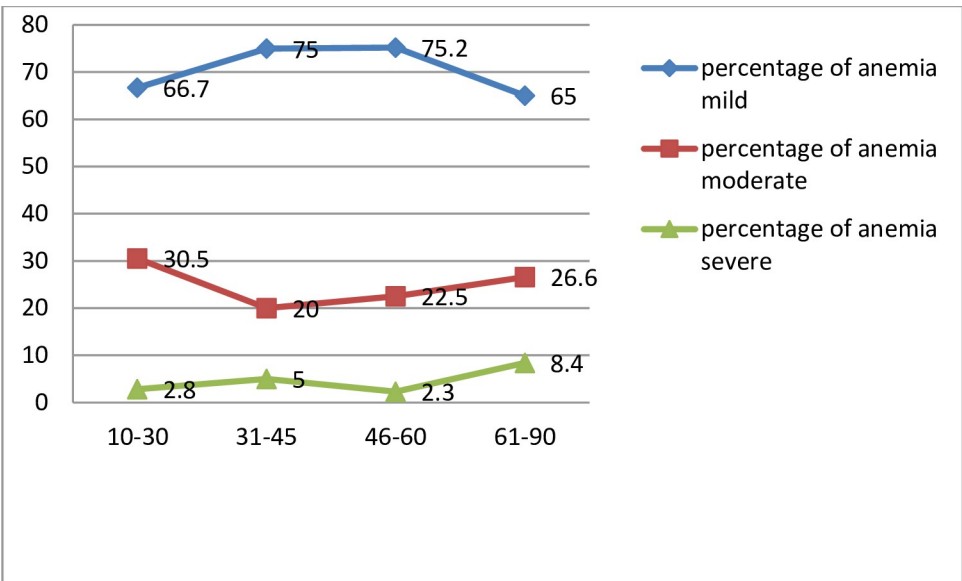

**Fig 2. Prevalence of mild, moderate and severe anemia in percentage by age.**

analysis and classification. In feature selection process, the features are elected based on their ranking and best suitability to the classifiers performance. Main methods of feature selection are (1) filter, (2) wrapper, and (3) embedded [38].

In this case, filter methods apply statistical measures to provide ranking scores for each feature. According to their score, they are either included or excluded from the dataset, examples include Chi-square and Gain ratio. Wrapper methods elect combinations feature subsets and assess their usability on a given machine learning algorithm. In view of that, the subsets are scored based on their predictive power. While, embedded methods learn the best features

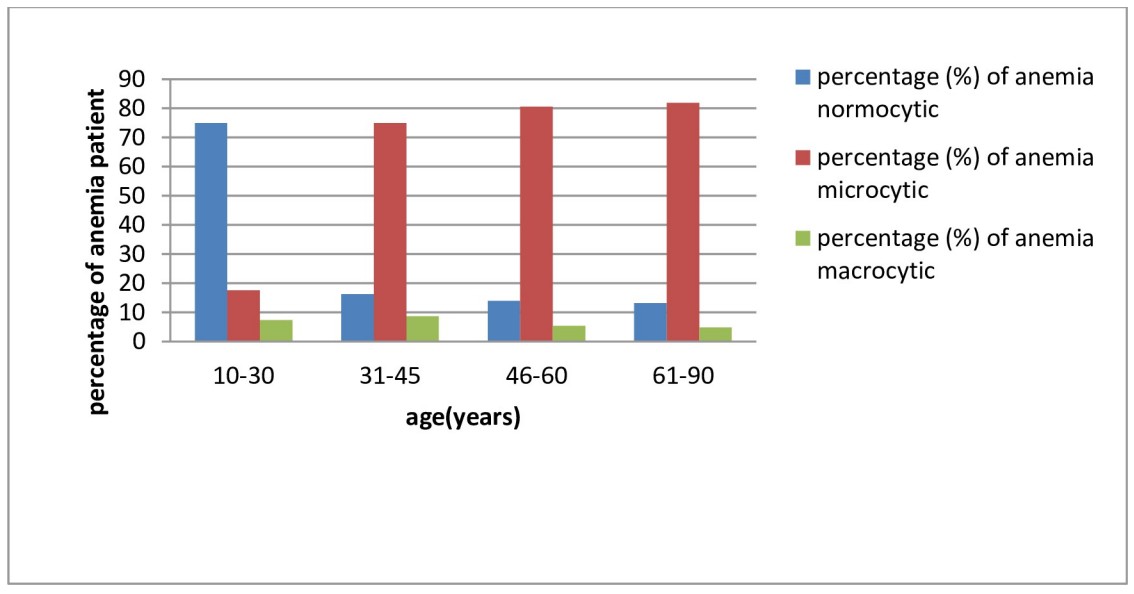

**Fig 3. Prevalence of normocytic, microcytic and macrocytic by age.**

**Table 4. Gender wise prevalence of Anemia in the study population.**

| | percentage (%) of anemia | | |
|---|---|---|---|
| | mild | moderate | severe |
| **Female** | 81.7734 | 14.28571429 | 3.9409 |
| **Male** | 55.90062 | 39.13043478 | 4.9689 |

according to the correctness (accuracy) of the learning model, Common types of embedded methods include decision tree algorithms such as CART, random forest and C4.5.

In this paper, the dataset collected was entered in Excel format for the eleven attributes mentioned in the Material and methods of Section 2.

The following techniques were used for our feature selection methods.

1. CART based feature selection: CART based feature selection technique is also a widely used feature selection technique due to its distinguishable capability of the features. This is a tree based classification algorithm that ranks the features based on their distinguishing capability and few features from the top would be selected based on the user need and criteria.

2. Correlation based feature selection: This is another popular feature selection technique that selects those features which are marginally co-related or not co-related, this is filter based feature selection and it is a multivariate method. In this case, if two or more highly correlated features exist, it tries to select only one feature among them.

3. Recursive feature selection: This is one of the prominent feature selection techniques that are available in Scikit-Learn. It recursively selects the important features in each training phase and discards the less important one.

Finally, we performed majority voting for feature selection on Recursive feature selection using gradient boosting, correlation and CART. Feature selection procedure is shown in the Fig 4.

Our analysis indicated that 7 attributes namely Age, Sex, PCV, MCH, MCHC, PLT and HGB are highly ranked. Hence, patient records with this reduced feature are used for our experiments.

## Work flow and framework of the proposed model

The original data collected is referred to as dataset 1. Then feature selection on dataset 1 was performed using majority voting which was applied using Recursive feature selection using gradient boosting, correlation and CART, we refer to this data as dataset 2. Dataset 1 was imbalanced, hence SMOTE technique was used to create dataset 3. We followed same

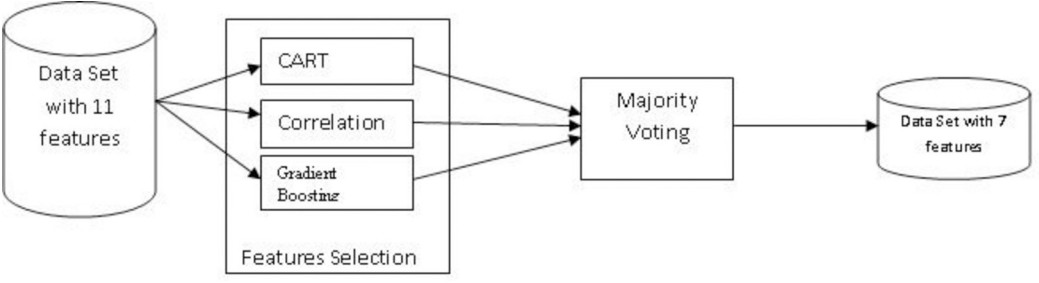

**Fig 4. Feature selection.**

procedure for feature selection on dataset3to create dataset 4. We applied six machine learning techniques for our 4 datasets and discussed results in the Results and Discussion section. Framework of proposed model shown in Fig 5.

## Techniques employed

The following six popular computational models, namely Support Vector Machine (SVM) [39–41], Random Forest (RF) [42], Multilayer perceptron (MLP) [43–45], Decision Tree (DT) [46, 47], Naïve Bayes (NB) [48] and Logistic Regression (LR) [49] were applied for anemia disease classification. These techniques were selected as they are well known, widely used and provide a framework of results for further research to be carried out for classification using ensemble and other models.

## Performance assessment parameters

Performance assessment is an important step for developing reliable and useful classifier. In our study we used five standard quality measure namely accuracy, sensitivity, specificity, area under the curve (AUC) and Kappa statistic for the performance assessment of the various classification techniques.

These are defined as follows:

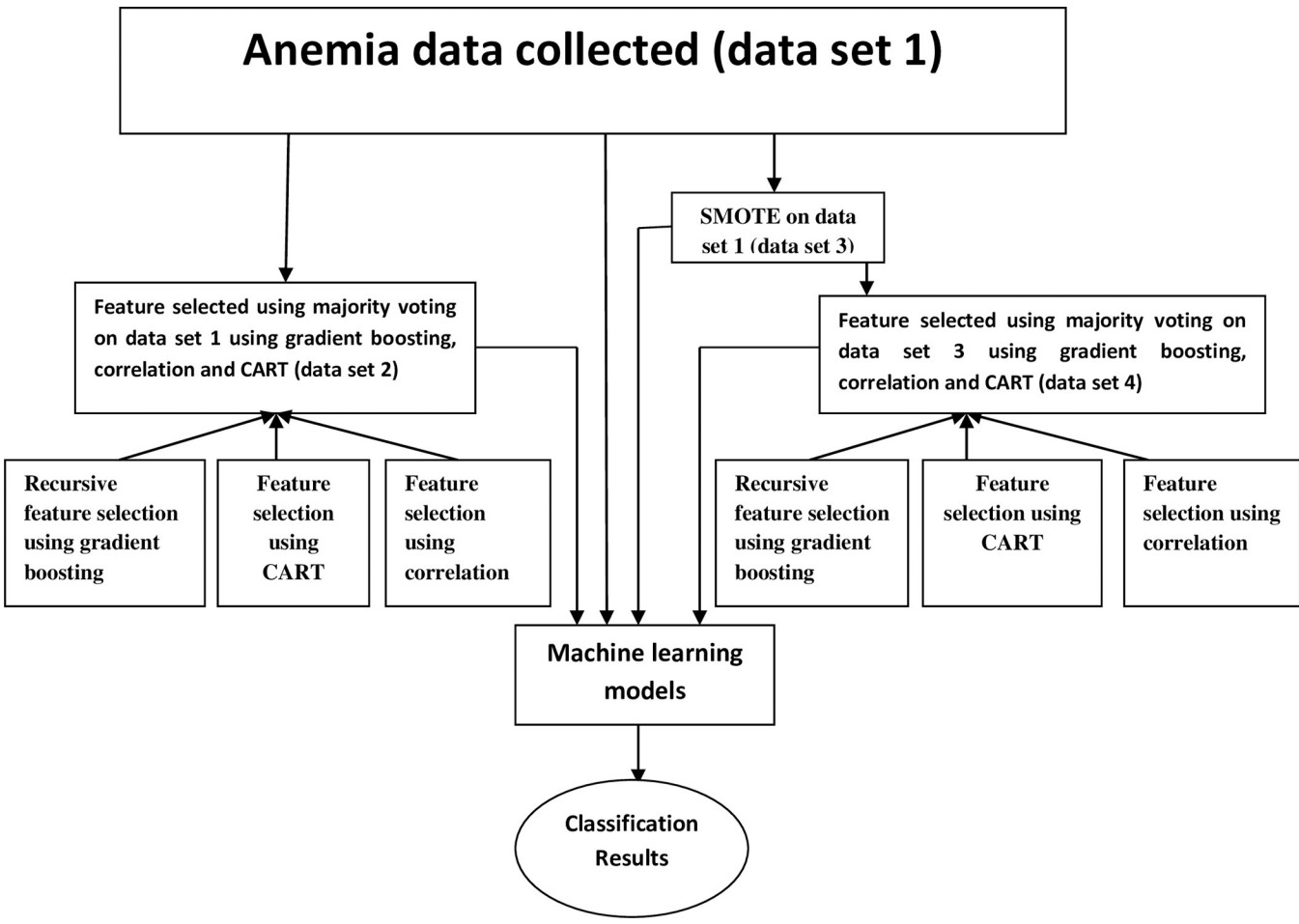

**Fig 5. Framework of the proposed model.**

| confusion matrix | | PREDICTED VALUES | | |
|---|---|---|---|---|
| ACTUAL VALUES | | **mild** | **moderate** | **severe** |
| | **Mild** | true mild | false moderate | false severe |
| | **moderate** | false mild | true moderate | false severe |
| | **severe** | false mild | false moderate | true severe |

Recall is the measure of proportion of the true positives, which are correctly identified.

$$Recall = \frac{True\ positive}{(True\ positive + False\ negative)}$$

For our problem recall defined for all the three class as follows:

$$Recall(mild) = \frac{true\ mild}{total\ mild(actual)}$$

$$Recall(moderate) = \frac{true\ moderate}{total\ moderate(actual)}$$

$$Recall(severe) = \frac{true\ severe}{total\ severe(actual)}$$

Precision is the measure of proportion of the true negatives, which are correctly identified. For our problem precision defined for all the three class as follows:

$$Precision = \frac{True\ positive}{(True\ positive + False\ positive)}$$

$$Precision(mild) = \frac{true\ mild}{total\ mild(pridicted)}$$

$$Precision(moderate) = \frac{true\ moderate}{total\ moderate(pridicted)}$$

$$Precision(severe) = \frac{true\ severe}{total\ severe(pridiced)}$$

Accuracy is the measure of proportion of true positives and true negatives, which are correctly identified.

$$Accuracy = \frac{(True\ positive + True\ negative)}{(True\ positive + True\ negative + False\ positive + False\ negative)}$$

AUC used in classification analysis in order to determine which of the used models predicts the classes best. The AUC lies between 0 and 1, where a perfect classifier can take a maximum value of 1.

The Kappa statistic is used to control only those instances that may have been correctly classified by chance. It is a good measure to handle multi class and imbalanced class problems. It is a measure of the agreement between the predicted and the actual classifications in a data set.

Note that Total accuracy is the observed agreement between the predicted and the actual classification in a data set [50].

Consider the following Binary classification confusion matrix:

| A\B | YES | NO |
|---|---|---|
| YES | a | b |
| NO | c | d |

The data on the main diagonal (a and d) represents the count of the number of agreements between the two trials A and B while the off diagonal data (b and c) represents the count of the number of disagreements between the two trials A and B.

The observed proportionate agreement represents the Total accuracy and is denoted by the formula:

Total accuracy = (a+d)/(a+b+c+d)

Random accuracy:

Assume we have two readers of the data–one for each trial.

We define $p_{yes}$ = (a+b) /(a+b+c+d). (a+c) /(a+b+c+d)

Similarly

$P_{no}$ = (c+d)/(a+b+c+d). (b+d)/(a+b+c+d)

The random agreement probability is the probability that the two readers (judges) in the two trials agreed on either yes or no i.e. $p_{yes}$ + $p_{no}$ also called as the Random accuracy for the computation.

Thus we get the computed Kappa statistic as given in the above formula

$$\text{Kappa} = \frac{(\text{total accuracy} - \text{random accuracy})}{(1 - \text{random accuracy})}$$

## Results and discussion

All the experiments have been conducted exhaustively by fine tuning the various hyper parameters of the proposed machine learning algorithms. The results have been tabulated based on the datasets used for experiments. The experiments have been carried out using hold-out and 10-fold cross validation. Based on the literature survey, it has been found that the proposed six machine learning techniques have superior performance in the context of similar type of problems. Moreover, from visualization of the dataset used, it has been found our data is having a mixture of categorical and continuous types of values and the techniques used in this paper had been giving promising results for this type of data. In hold-out method the ratio of the training and testing taken are 80%and 20% respectively.

Table 5 represents the results of the various machine learning techniques used which has been tested on the original dataset. Whereas Table 6 represents the results of various machine learning techniques used which has been tested on our feature selected dataset. It can be observed from the Table 5 that logistic regression has given the highest accuracy and Kappa value for the hold-out and 10-fold cross validation methods. Whereas in the case of other metrics, like recall, precision and AUC based on the dataset and methodology used for experiment, different techniques had given promising results.

For Mild class as shown in Table 5, MLP had given the best recall values with 98% and the technique provides AUC values of 99.3 and 98.9 using hold-out and 10-fold cross validation methods, respectively. Similar case can be observed with Moderate class, MLP network has

**Table 5. Summary of the results obtained using the dataset1 (Anemia original dataset).**

| Class | | Algorithms | | | | | | | | | | | |
|---|---|---|---|---|---|---|---|---|---|---|---|---|---|
| | | DT | LR | MLP | NB | RF | SVM | DT | LR | MLP | NB | RF | SVM |
| | Experiment Method | Hold Out | | | | | | 10 fold cross validation | | | | | |
| | Accuracy (%) | 84.72 | **94.44** | 94.44 | 87.5 | 88.88 | 88.88 | 89.01 | **92.85** | 92.58 | 82.96 | 92.03 | 85.71 |
| | Kappa | 0.675 | **0.873** | 0.871 | 0.718 | 0.748 | 0.726 | 0.749 | **0.836** | 0.827 | 0.614 | 0.816 | 0.633 |
| Mild | Recall | 0.902 | 0.961 | **0.980** | 0.902 | 0.922 | 0.980 | 0.949 | 0.973 | **0.980** | 0.906 | 0.965 | 0.992 |
| | Precision | 0.979 | 0.980 | **1.000** | 0.939 | 0.959 | 0.926 | 0.953 | 0.958 | **0.973** | 0.913 | 0.961 | 0.879 |
| | AUC | 0.913 | 0.978 | **0.993** | 0.943 | 0.974 | 0.896 | 0.933 | 0.985 | **0.989** | 0.949 | 0.988 | 0.835 |
| Moderate | Recall | 0.722 | 0.944 | **1.000** | 0.833 | 0.889 | 0.778 | 0.804 | 0.826 | **0.902** | 0.685 | 0.870 | 0.620 |
| | Precision | 0.722 | 0.850 | **0.818** | 0.714 | 0.727 | 0.778 | 0.771 | 0.884 | **0.822** | 0.663 | 0.825 | 0.770 |
| | AUC | 0.725 | 0.958 | **0.985** | 0.926 | 0.953 | 0.852 | 0.881 | 0.944 | **0.976** | 0.891 | 0.970 | 0.779 |
| Severe | Recall | 0.667 | 0.667 | 0.000 | **0.667** | 0.333 | 0.000 | 0.483 | **0.813** | 0.188 | 0.438 | 0.500 | 0.063 |
| | Precision | 0.286 | 1.00 | 0.000 | **1.000** | 1.000 | 0.000 | 0.538 | **0.722** | 0.600 | 0.467 | 0.800 | 1.000 |
| | AUC | 0.797 | 0.986 | 1.000 | **0.995** | 0.995 | 0.882 | 0.849 | **0.993** | 0.986 | 0.903 | 0.978 | 0.870 |

*Abbrs: Decision Tree (DT), Logistic Regression (LR), Multilayer perceptron (MLP), NaïveBayes (NB), Random Forest (RF), Support Vector Machine (SVM)

dominated with recall values of 100 percent and 90.2 percent in the case of hold-out and 10-fold cross validation methods and AUC values of 98.5 and 97.6% in the case of hold-out and 10-fold cross validation methods. Severe class being the critical stage of the Anemia disease and due to the low quantity of samples of that data, it has yielded poor results.

The results obtained using the feature selected dataset in Table 6 show that in the case of Mild class, Decision Tree had given the best result with recall value of 91.5 percent, precision value of 95.6 percent and AUC value of 96.6 percent when hold-out method is used. Whereas in the case of 10-fold cross validation method, LR has given the recall value of 92.2 percent, precision value of 94.8 percent and AUC value of 99.0 percent. For the Moderate class, Random Forest had given the best result with recall value of 98.2 percent, precision value of 90.3

**Table 6. Summary of the results obtained using the dataset2 (feature selected).**

| Class | | Algorithms | | | | | | | | | | | |
|---|---|---|---|---|---|---|---|---|---|---|---|---|---|
| | | DT | LR | MLP | NB | RF | SVM | DT | LR | MLP | NB | RF | SVM |
| | Experiment Method | Hold Out | | | | | | 10 fold cross validation | | | | | |
| | Accuracy (%) | **96.10** | 94.80 | 95.45 | 85.06 | 95.45 | 90.90 | 91.27 | 95.05 | **95.31** | 84.11 | 94.01 | 89.97 |
| | Kappa | **0.941** | 0.921 | 0.931 | 0.774 | 0.931 | 0.862 | 0.869 | 0.925 | **0.929** | 0.761 | 0.910 | 0.849 |
| Mild | Recall | **0.915** | 0.894 | 0.872 | 0.851 | 0.872 | 0.830 | 0.918 | **0.922** | 0.914 | 0.844 | 0.926 | 0.828 |
| | Precision | **0.956** | 0.933 | 0.976 | 0.851 | 0.976 | 0.975 | 0.929 | **0.948** | 0.955 | 0.893 | 0.948 | 0.972 |
| | AUC | **0.966** | 0.988 | 0.993 | 0.968 | 0.998 | 0.965 | 0.970 | **0.990** | 0.991 | 0.969 | 0.994 | 0.965 |
| Moderate | Recall | 0.965 | 0.947 | 0.982 | 0.825 | **0.982** | 0.930 | 0.863 | 0.930 | **0.949** | 0.797 | 0.902 | 0.887 |
| | Precision | 0.932 | 0.915 | 0.903 | 0.783 | **0.903** | 0.841 | 0.877 | 0.926 | **0.917** | 0.745 | 0.917 | 0.825 |
| | AUC | 0.962 | 0.988 | 0.992 | 0.913 | **0.998** | 0.913 | 0.944 | 0.987 | **0.989** | 0.908 | 0.988 | 0.896 |
| Severe | Recall | **1.000** | 1.000 | 1.000 | 0.880 | 1.000 | 0.960 | 0.957 | **1.000** | 0.996 | 0.883 | 0.992 | 0.984 |
| | Precision | **1.000** | 1.000 | 1.000 | 0.936 | 1.000 | 0.941 | 0.932 | **0.977** | 0.988 | 0.897 | 0.955 | 0.916 |
| | AUC | **1.000** | 1.000 | 1.000 | 0.979 | 1.000 | 0.982 | 0.975 | **0.998** | 0.997 | 0.978 | 0.999 | 0.976 |

*Abbrs: Decision Tree (DT), Logistic Regression (LR), Multilayer perceptron (MLP), Naïve Bayes (NB), Random Forest (RF), Support Vector Machine (SVM)

**Table 7. Summary of the results obtained using the dataset3 (SMOTE).**

| Class | | Algorithms | | | | | | | | | | | |
|---|---|---|---|---|---|---|---|---|---|---|---|---|---|
| | | DT | LR | MLP | NB | RF | SVM | DT | LR | MLP | NB | RF | SVM |
| | Experiment Method | Hold Out | | | | | | 10 fold cross validation | | | | | |
| | Accuracy (%) | 94.80 | 96.75 | **99.35** | 85.71 | 97.40 | 93.50 | 89.21 | 92.10 | **94.21** | 82.63 | 92.36 | 86.05 |
| | Kappa | 0.921 | 0.951 | **0.990** | 0.784 | 0.960 | 0.902 | 0.774 | 0.834 | **0.879** | 0.643 | 0.84 | 0.681 |
| Mild | Recall | 0.894 | 0.957 | **0.979** | 0.894 | 0.957 | 0.894 | 0.953 | 0.965 | **0.977** | 0.898 | 0.969 | 0.988 |
| | Precision | 0.955 | 0.938 | **1.000** | 0.894 | 0.957 | 0.933 | 0.953 | 0.950 | **0.969** | 0.920 | 0.958 | 0.878 |
| | AUC | 0.980 | 0.988 | **0.992** | 0.980 | 0.997 | 0.964 | 0.944 | 0.983 | **0.992** | 0.955 | 0.988 | 0.855 |
| Moderate | Recall | 0.965 | 0.947 | **1.000** | 0.825 | 0.965 | 0.912 | 0.804 | 0.804 | **0.848** | 0.674 | 0.826 | 0.620 |
| | Precision | 0.902 | 0.964 | **0.983** | 0.797 | 0.965 | 0.912 | 0.763 | 0.860 | **0.907** | 0.639 | 0.854 | 0.760 |
| | AUC | 0.971 | 0.989 | **0.992** | 0.916 | 0.997 | 0.930 | 0.913 | 0.937 | **0.981** | 0.889 | 0.971 | 0.779 |
| Severe | Recall | 0.980 | 1.000 | **1.000** | 0.860 | 1.000 | 1.000 | 0.656 | 0.906 | **0.938** | 0.688 | 0.844 | 0.531 |
| | Precision | 1.000 | 1.000 | **1.000** | 0.896 | 1.000 | 0.962 | 0.778 | 0.853 | **0.833** | 0.667 | 0.844 | 1.000 |
| | AUC | 0.990 | 1.000 | **1.000** | 0.977 | 1.000 | 0.990 | 0.898 | 0.993 | **0.996** | 0.975 | 0.994 | 0.991 |

*Abbrs: Decision Tree (DT), Logistic Regression (LR), Multilayer perceptron (MLP), NaïveBayes (NB), Random Forest (RF), Support Vector Machine (SVM)

and AUC value of 99.8 percent when tested using hold-out method. Whereas in the case of10-fold cross validation Multilayer Perceptron (MLP) had given the top results with a recall value of 94.9, precision value of 91.7 and AUC value of 98.9. It can be observed that the results of the Severe class have been totally different from the earlier two classes in case of hold-out method and 10-fold cross validation methods. In the case of the hold-out, Decision Tree (DT), Logistic Regression (LR), MLP and RF have given good recall, precision and AUC values of 100 percent whereas in the case of 10-fold cross validation Logistic regression has given the high recall value of 100 percent, precision of 97.7 and AUC value of 99.8 percent.

As the dataset is having disproportioned samples for the three classes, we utilized Synthetic Minority Oversampling Technique (SMOTE) to balance our dataset. Tables 7 and 8 provide the results of the experiments obtained using the SMOTE dataset. Table 7 indicates that MLP

**Table 8. Summary of the results obtained using the dataset4 (SMOTE and feature selected).**

| Class | | Algorithms | | | | | | | | | | | |
|---|---|---|---|---|---|---|---|---|---|---|---|---|---|
| | | DT | LR | MLP | NB | RF | SVM | DT | LR | MLP | NB | RF | SVM |
| | Experiment Method | Hold Out | | | | | | 10 fold cross validation | | | | | |
| | Accuracy (%) | 88.88 | **94.44** | 88.88 | 87.5 | 88.88 | 81.94 | 86.26 | **91.48** | 90.38 | 82.41 | 89.01 | 82.41 |
| | Kappa | 0.76 | **0.870** | 0.734 | 0.710 | 0.748 | 0.517 | 0.685 | **0.801** | 0.778 | 0.589 | 0.747 | 0.535 |
| Mild | Recall | 0.902 | **0.980** | 0.961 | 0.922 | 0.922 | 0.980 | 0.934 | **0.969** | 0.953 | 0.910 | 0.949 | 0.988 |
| | Precision | 1.000 | **0.980** | 0.942 | 0.922 | 0.959 | 0.847 | 0.934 | **0.943** | 0.957 | 0.886 | 0.946 | 0.849 |
| | AUC | 0.944 | **0.973** | 0.968 | 0.927 | 0.974 | 0.771 | 0.927 | **0.979** | 0.976 | 0.941 | 0.978 | 0.785 |
| Moderate | Recall | 0.944 | **0.944** | 0.833 | 0.778 | 0.889 | 0.500 | 0.750 | 0.815 | **0.880** | 0.641 | 0.804 | 0.511 |
| | Precision | 0.708 | **0.850** | 0.750 | 0.737 | 0.727 | 0.692 | 0.719 | 0.852 | **0.771** | 0.663 | 0.771 | 0.712 |
| | AUC | 0.942 | **0.975** | 0.965 | 0.900 | 0.964 | 0.713 | 0.870 | 0.959 | **0.965** | 0.899 | 0.956 | 0.721 |
| Severe | Recall | **0.889** | 0.333 | 0.000 | 0.667 | 0.333 | 0.000 | 0.375 | 0.625 | 0.250 | **0.500** | 0.438 | 0.000 |
| | Precision | **0.906** | 1.000 | 0.000 | 1.000 | 1.000 | 0.000 | 0.500 | 0.769 | 1.000 | **0.667** | 0.636 | 0.000 |
| | AUC | **0.969** | 1.000 | 0.969 | 0.995 | 0.969 | 0.752 | 0.818 | 0.990 | 0.995 | **0.973** | 0.974 | 0.764 |

*Abbrs: Decision Tree (DT), Logistic Regression (LR), Multilayer perceptron (MLP), Naïve Bayes(NB), Random Forest (RF), Support Vector Machine (SVM)

has out-performed all other techniques with the highest Accuracy of 99.35 percent in the case of hold-out method and a value of 94.21 percent in the case of 10 fold cross validation method. The Kappa statistic has shown the advantages of MLP network with values of 0.99 and 0.879 using hold-out and 10-fold cross validation methods. Looking at the recall and AUC values in the case of Mild class using the hold-out method, the MLP provided recall value of 97.9 percent and AUC value of 99.2 percent. When the 10-fold cross validation method is used for the same class, the MLP network showed recall value of 97.7 percent and AUC value of 99.2 percent. For the Moderate class the MLP has given the recall value of 100.0 percent and AUC value of 99.2 using hold-out method, and a recall value of 84.8 percent and AUC value of 98.1 in the case of 10-fold cross validation. For the Severe class, as it can be shown in Table 7, the MLP, LR and RF have given competitively 100 percent recall and AUC when tested using hold-out method. The reason behind getting so overfitting results across different ML techniques in this case can be attributed to the synthetic values generated by the SMOTE technique which had a close resemblance to the actual values and most of the ML techniques tend to over fit. In the case of 10-fold cross validation method the only MLP has given a best recall and AUC values of 93.8 percent and 99.6 percent.

Table 8 describes the results using SMOTE and feature selected dataset. It is evident from this table that Decision Tree (DT) has performed well for the severe class in the case of hold-out method and for the same class in the case of 10-fold cross validation method Naïve Bayes had given the recall value of 50.0 percent and AUC value of 97.3 percent. As far as Mild class is concerned LR has topped in both the cases of holdout method and 10-fold cross validation method. The recall and AUC values of 98.0 and 97.3 are obtained when holdout method is used and in the case of 10-fold cross validation it was recall value of 96.9 and AUC value of 97.9 percent. When Moderate case is considered again LR has given best results with recall value of 94.4 percent and AUC value of 97.5 percent when run using holdout method. Whereas in the case of 10-fold cross validation method for the same class the MLP has topped with recall value of 88.0 percent and AUC value of 96.5 percent. Similarly when Accuracy and Kappa values are considered, irrespective of the method used, LR has topped.

The reason for getting good results with this dataset can be attributed to the class balanced property which has been achieved with the help of SMOTE technique used and due to the feature selection method adopted for the dataset after applying SMOTE, has filtered out the noise in the dataset and had improvised the classification capabilities of the machine learning algorithms.

As far as the reason regarding performance of the machine learning algorithms in Table 5 is concerned MLP was predominant among all the techniques used, this can be attributed to the inherent nature of MLP as being a neural network, it was good at capturing the learning capabilities from the data for the majority classes. As severe class was not having enough samples to train a model well, MLP failed in that class. LR being good model for predicting the continuous values, it had performed better in case of severe class as the dataset was also having enough ratio of continuous values.

When we look at Table 6 results, it can be observed that DT and RF had given competitively good performance over other methods in the case of holdout method. This can be attributed due to the reduction of noise when feature selection is done. Whereas in the case of 10-fold cross validation LR and MLP were competitively performing well in mutually opposite classes compared with other methods. In the case of Table 7, since the dataset is balanced using SMOTE method, the learning features of the dataset had increased thereby giving a good scope for neural network based MLP technique. Hence, MLP has unanimously dominated over other algorithms irrespective of mode of training (i.e., Hold-out or 10-fold cross validation)

Based on the given classes in the dataset and their significance in the field of medical domain, we can summarize the results as follows:

- Mild class being the starting stage of the Anemia, it's crucial to identify patients who have been diagnosed in this stage and inform them as early as possible so that precautionary measures can be taken to ensure that their Anemia does not deteriorate rapidly. Hence, the MLP helps in identifying the Mild type of cases more efficiently, as it has given a good recall value for this class in most of the cases.

- Moderate class is the middle stage of the Anemia, where most of the patients could have reached unknowingly and have lesser number of patients compared to the Mild class. And here most of the patients get aware of the Anemia symptoms and can change their lifestyle for prolonging their lifetime. Again, MLP helps to identify patients with Moderate cases more accurately, since it has provided good recall values.

- Severe class being the most advanced stage of Anemia, it is of least significance, since the patient would have become aware of this disease by this stage and must have started the treatment course toward shading the Anemia. For identifying whether the patient has reached this stage or not, the Decision Tree and MLP would help us, as they have given good recall values in most of the cases.

## Conclusions and directions for further work

Anemia is one of the prevalent diseases among women and children globally, the disease needs to be identified and treated in its earlier stages since it can affect academic performance and work output of adults thus affecting a nation's economy and society. In this line, this paper addresses the problem of identifying the Anemia disease at various stages with the help of different machine learning techniques that can accurately classify the patient to the class (stage of Anemia). Mild-class (first stage) being the critical stage where in identifying and avoiding further advancement into deteriorating stages can be done. MLP showed promising results for the classification of Anemia. For the Moderate class, MLP demonstrates best recall and AUC values. The Severe class being the most advanced stage of Anemia, has least significance in classification, as most of the patients would have been informed about the existence of this disease by this stage. Yet, there might be few cases, where the patient would not be knowing about the existence of Anemia at this stage. In this context, this paper tries to predict the existence of Anemia at this stage accurately with the help of Decision Tree and MLP. In summary, the paper tries to predict the existence of the Anemia at various stages with the help of Machine Learning techniques that have proven to be accurate for our Anemia dataset.

Data was collected in an outpatient clinical setting in India and the prevalence of anemia by age and gender were computed. The study used three feature selection techniques with majority voting to identify the most significant features in the input data set. Various multi class classification algorithms were used to perform the diagnosis of anemia as mild, moderate and severe. SMOTE techniques were used to deal with the class imbalance problem since the original data set was imbalanced. In all four data sets including the original data set were used to perform the data experiments. Performance benchmarking for the six machine learning algorithms used was done and tabulated using both the 10-fold cross validation and hold out methods.

A comparative analysis of the classification results obtained can be done by sourcing the Anemia patient data set from other locations and regions. It is planned to source this data set from Africa in the near future so that a comparative study of the results obtained can be performed.

Furthermore, using machine learning techniques to rank possible social determinants of anemia can be performed. The study can also be localised to a community setting whereby macro health indices relating to diet and nutrition can be computed. Accordingly, such interventions can be designed on the basis of the computed macro health indices which relate both to the nutrition and diet side of policy setting as well as other macro indices in the community such as financial indices and the broader socio-economic environment.

The computing of nutrition related indices like Diet diversity score, Food security score and malnutrition score can be used to design nutrition related public health interventions. The distribution of these scores in the community can help us design targeted interventions for the benefit of the community. In addition, scores like wealth index give an idea of the financial status of the community residents. The distribution of this score can help in designing policy measures on the social welfare side of policy planning. These computations can also help in designing suitable aggregated social and health score cards for the community.

Finally, a new hybrid model can be designed for classifying iron deficiency anemia that is based on the concepts of Deep learning, genetic algorithms and convolutional neural networks (CNN) using the same data sets.

## Supporting information

**S1 Appendix.**
(DOCX)

## Acknowledgments

The authors would like to thank the Eureka diagnostic center, Lucknow, India that helped to collect and analyze patient data for the CBC test reports.

## Author Contributions

**Conceptualization:** Rajan Vohra, Anil Kumar Dudyala, Jankisharan Pahareeya.

**Data curation:** Wasiq Khan.

**Formal analysis:** Rajan Vohra, Anil Kumar Dudyala, Jankisharan Pahareeya.

**Investigation:** Rajan Vohra.

**Methodology:** Rajan Vohra, Jankisharan Pahareeya.

**Supervision:** Rajan Vohra, Abir Hussain.

**Writing – original draft:** Rajan Vohra.

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
