## [Decision Letter · Decision Letter 0]

1 Nov 2021

PONE-D-21-28346Multi class classification algorithms for the diagnosis of Anemia in an outpatient clinical setting.PLOS ONE

Dear Dr. Vohra,

Thank you for submitting your manuscript to PLOS ONE. After careful consideration, we feel that it has merit but does not fully meet PLOS ONE’s publication criteria as it currently stands. Therefore, we invite you to submit a revised version of the manuscript that addresses the points raised during the review process.

We look forward to receiving your revised manuscript.

Kind regards,

Wajid Mumtaz

Academic Editor

PLOS ONE

Journal Requirements:

3. Please include your tables as part of your main manuscript and remove the individual files. Please note that supplementary tables (should remain/ be uploaded) as separate "supporting information" files

5. Please include your full ethics statement in the ‘Methods’ section of your manuscript file. In your statement, please include the full name of the IRB or ethics committee who approved or waived your study, as well as whether or not you obtained informed written or verbal consent. If consent was waived for your study, please include this information in your statement as well

Reviewers' comments:

Reviewer's Responses to Questions

**Comments to the Author**

1. Is the manuscript technically sound, and do the data support the conclusions?

Reviewer #1: Partly

2. Has the statistical analysis been performed appropriately and rigorously? 

Reviewer #1: I Don't Know

3. Have the authors made all data underlying the findings in their manuscript fully available?

Reviewer #1: Yes

4. Is the manuscript presented in an intelligible fashion and written in standard English?

Reviewer #1: Yes

5. Review Comments to the Author

Reviewer #1: In this manuscript, the authors proposed a multiclass classification algorithm using six machine learning techniques for diagnosing anemia severity. This paper diagnoses the problem of multi-class (three classes) classification of mild, moderate, and severe anemia. Experiments were performed with a hold out method and 10 fold cross validation for multi-class classification using 6 standard machine learning algorithms using 4 data sets including the original data set. However, for a more complete manuscript, I present a few supplements and fixes.

Major comment

1. On page 10, In line 2 of the 1th paragraph Data Analysis, More explanations are needed for feature selection techniques.

2. On page 13, In line 3-6 of the 2th paragraph Results and Discussion, LR is correct to achieve the highest accuracy and kappa value in table 5, but DT and MLP are the highest values in table 6, respectively. The sentence should be modified to fit the table.

3. On page 13, In Results and Discussion, the contents of the 4th paragraph do not match Table6.

4. On page 13, In the Results and Discussion, There seems to be no clinical meaning in the line 11-13 of the 5th paragraph. Since the severe class is a relatively small percentage of the data set, when a smote is used, the overfitting result is natural. In addition to MLP, other algorithms also achieved 100%, so it does not seem to be an ideal number.

5. On page 13, In Results and Discussion, the contents of the 6th paragraph do not match Table8.

6. On page 14, In Results and Discussion, lines 1 and 2 of the 9th paragraph do not match Table6.

7. The criteria for classifying mild, moderate, and severe are not mentioned in the manuscript.

Minor comment

1. It would be better if the designation of anemia and anaemia were unified in the manuscript.

2. On page 7, In line 5 of paragraph Abstract, Rather than a simple description of dividing into three classes, it seems necessary to explain more about the reason for dividing.

3. On page 7, In line 3 of paragraph Introduction, In front of ‘Anaemiaprevalence’, ‘,’ is required and spaces should be written.

4. On page 7, In line 4 of paragraph Introduction, A space is needed in ‘WhileAnaemia’.

5. On page 7, In line 10 of paragraph Introduction, A space is needed in ‘Anaemiasis’.

6. On page 9, In line 1 of the 7th paragraph Introduction, A space is needed in ‘analysiswhile’.

7. On page 10, In line 9 from the bottom of the paragraph Data Analysis, A space is needed in ‘Table 3shows.

8. On page 12, It seems that Precision(moderate) and Precision(severe) should be changed to recall(moderate), recall(severe).

9. On page 12, It seems that true negative should be defined as true positive in precision formula.

10. On page 12, The exact definition or explanation of total accuracy and random accuracy in the formula for kappa should be included in the text.

11. On pages 23 to 25, In Table 5, Table 6, Table 7, and Table 8, it seems to indicate (%) in Accuracy.

12. On page 24, In the title of Table 7, it seems that Smote should be capitalized as SMOTE.

13. On page 36, In the title of Table 8, 'Smote fs' seems to be written as 'SMOTE and feature selected'. If it is written as an abbreviation such as 'fs', it may not be understood by readers

14. On page 20, Since the contents of Fig 2 and Table2 overlap, it would be better to use only one of them.

6. PLOS authors have the option to publish the peer review history of their article (what does this mean?). If published, this will include your full peer review and any attached files.

Reviewer #1: No

---

## [Author Response · Author response to Decision Letter 0]

22 Dec 2021

The authors would like to first and foremost thank the Editor, PLOS ONE and the reviewer, for taking out time and putting efforts in finding the shortcomings of the article and giving their supportive comments in making the article strong.

 The comments/suggestions given by the reviewer are addressed one by one in the following section to the full of our knowledge and belief. 

The following are the comments/suggestions given by the reviewer #1.

Major Comments:

1. On page 4, in line 2 of the 1th paragraph Data Analysis, More explanations are needed for feature selection techniques.

Resolution: The authors have provided detailed information about feature selection techniques and their techniques used for this research works.

2. On page 10, in line3-6 of the 2th paragraph Results and Discussion, LR is correct to achieve the highest accuracy and kappa value in table 5, but DT and MLP are the highest values in table 6 respectively. The sentence should be modified to fit the table.

Resolution: The author would like to thank reviewer for pointing out the mistake. It has been corrected.

3. On page 10, in Results and Discussion, the contents of the 4th paragraph do not match Table 6.

Resolution: The contents of the 4th paragraph has been updated according to the Table 6 values.

4. On page 10, in the Results and Discussion, there seems to be no clinical meaning in the line11-13 of the 5th paragraph. Since the severe class is relatively small percentage of the dataset, when a SMOTE is used, the overfitting result is natural. In addition to MLP, other algorithms also achieved 100%, so it does not seem to be an ideal number.

Resolution: The authors agree with the reviewers comments. The reason behind getting this type of overfitting results is justified in the subsequent lines 13-15 of the said paragraph.

5. On page 10, in Results and Discussion, the contents of the 6th paragraph do not match Table 8.

Resolution: The contents of the 6th paragraph of the Results and Discussion has been updated according to Table 8.

6. On page 11, in Results and Discussion, lines 1 and 2 of the 9th paragraph do not match Table 6.

Resolution: Relevant correction has been done as per the Table 6 to rectify the mistake made when writing the 9th paragraph of Results and Discussion.

7. The criteria for classifying mild, moderate, and severe are not mentioned in the manuscript.

Resolution:The criteria for classifying an anemia into mild, moderate and severe is given at 1st paragraph On page 4, of the Data Analysis.

Minor Comments:

1. It would be better if the designation of anemia and anaemia were unifield in the manuscript

Resolution: keyword anaemia is replaced with anemia in the manuscript.

2. On the page 1, in line 5 of paragraph abstract, rather than a simple description of dividing in to three classes, it seems necessary to explain more about the reason for dividing.

Resolution:The criteria for classifying an anemia into mild, moderate and severe is given at 1st paragraph of the Data Analysis.

3. On page 1,In line 3 of paragraph Introduction,in front of ‘Anaemiaprevalence’, ‘,’ isrequired and space should be written.

Resolution: Done

4. On the page 1,In line 4 of paragraph Introduction,A space is needed in ‘whileanaemia’.

Resolution: Done

5. On the page 1,In line 10 of paragraph Introduction,Aspace is needed in ‘Anaemiasis’.

Resolution: Done

6. On the page 2,In line 1 of the 7th paragraph Introduction,A space is needed in ‘analysiswhile’.

Resolution: Done

7. On the page 5, In line 9from the bottom of the paragraph Data Analysis, A space is needed in ‘Table 3shows’.

Resolution: Done

8. On page 8, It seems that Precision (Moderate) and Precision (severe) should be changed to recall (moderate).recall (severe).

Resolution: corrected 

9. On page 8,It seems that true negative should be defined as true positive in precision formula

Resolution: corrected 

10. On page 9, the exact definition or explanation of total accuracy and random accuracy in the formula for kappa should be included in the text.

Resolution: Done

11. On pages 12 to 14,In Table 5, Table 6, Table 7, and Table 8, it seems to indicate(%) in Accuracy.

Resolution: Done

12. On page 13, in the title of Table 7, it seems that Smote should be capitalized as SMOTE.

Resolution: Done

13. On page 14, in the title of table 8, ‘Smote fs’ seems to be written as SMOTE and feature selected’. If it is written as an abbreviation such ‘fs’, it may not be understood by readers. 

Resolution: Done

14. On page 5, since the contents of fig 2 and Table 2 overlap, it would be better to use only one of them.

Resolution: Done

---

## [Decision Letter · Decision Letter 1]

15 Mar 2022

PONE-D-21-28346R1Multi class classification algorithms for the diagnosis of Anemia in an outpatient clinical setting.PLOS ONE

Dear Dr. Rajan Vohra,

Thank you for submitting your manuscript to PLOS ONE. After careful consideration, we feel that it has merit but does not fully meet PLOS ONE’s publication criteria as it currently stands. Therefore, we invite you to submit a revised version of the manuscript that addresses the points raised during the review process.

We look forward to receiving your revised manuscript.

Kind regards,

Wajid Mumtaz

Academic Editor

PLOS ONE

Journal Requirements:

Reviewers' comments:

Reviewer's Responses to Questions

**Comments to the Author**

1. If the authors have adequately addressed your comments raised in a previous round of review and you feel that this manuscript is now acceptable for publication, you may indicate that here to bypass the “Comments to the Author” section, enter your conflict of interest statement in the “Confidential to Editor” section, and submit your "Accept" recommendation.

Reviewer #1: (No Response)

2. Is the manuscript technically sound, and do the data support the conclusions?

Reviewer #1: Yes

3. Has the statistical analysis been performed appropriately and rigorously? 

Reviewer #1: Yes

4. Have the authors made all data underlying the findings in their manuscript fully available?

Reviewer #1: Yes

5. Is the manuscript presented in an intelligible fashion and written in standard English?

Reviewer #1: Yes

6. Review Comments to the Author

Reviewer #1: Referring to review, thank you for revising the manuscript. I have reviewed your revised version. The overall composition has become intuitive to make it easier for the reader to understand, but I have found some modifications. I'm leaving a few supplements and fixes.

Major comment

1. On page 11, In line 2-4 of the 4th paragraph Data Analysis, In Figure 2, the percentage of anemia mild is 66.8% in the 10-30 years age group and 75% in the 31-45 years age group. However, when checked in Figure 1, the height of the bar graph does not fit, so it seems necessary to modify it.

Minor comment

1. On page 11, In line 6-8 of the 4th paragraph Data Analysis, if there is a clinical reason for mentioning the lowest prevalent age group, and the second most prevalent age group for each class, it seems better to write down the reason. If you don't have any special reason, it seems good to mention only the lowest prevalent age group.

2. On page 14, In line 1-3 of the first paragraph Workflow and Framework of the proposed model, all data sets are mentioned, but Figure 5 shows only data sets1 and 3 separately. For readers' intuitive understanding, it seems better to add a notation for data set 2 and data set 4 to Figure 5.

3. On page 14, In line 4 of the first paragraph Workflow and Framework of the proposed model, the result of applying feature selection to data set 3 is data set 4, so you need to modify the number.

4. On page 15, It seems that true negative of the denominator should be defined as true positive in precision formula.

5. On page 16, In line 11 of the 4th paragraph, it seems that AUV should be modified to AUC.

7. PLOS authors have the option to publish the peer review history of their article (what does this mean?). If published, this will include your full peer review and any attached files.

Reviewer #1: No

---

## [Author Response · Author response to Decision Letter 1]

7 Apr 2022

All review comments made by the reviewer have been done and the manuscript has been updated accordingly and actions taken listed in the rebuttal letter attached.

---

## [Decision Letter · Decision Letter 2]

26 May 2022

Multi class classification algorithms for the diagnosis of Anemia in an outpatient clinical setting.

PONE-D-21-28346R2

Dear Dr. Rajan Vohra,

We’re pleased to inform you that your manuscript has been judged scientifically suitable for publication and will be formally accepted for publication once it meets all outstanding technical requirements.

Kind regards,

Wajid Mumtaz

Academic Editor

PLOS ONE

Additional Editor Comments (optional):

Reviewers' comments:

Reviewer's Responses to Questions

**Comments to the Author**

1. If the authors have adequately addressed your comments raised in a previous round of review and you feel that this manuscript is now acceptable for publication, you may indicate that here to bypass the “Comments to the Author” section, enter your conflict of interest statement in the “Confidential to Editor” section, and submit your "Accept" recommendation.

Reviewer #1: All comments have been addressed

2. Is the manuscript technically sound, and do the data support the conclusions?

Reviewer #1: Yes

3. Has the statistical analysis been performed appropriately and rigorously? 

Reviewer #1: Yes

4. Have the authors made all data underlying the findings in their manuscript fully available?

Reviewer #1: Yes

5. Is the manuscript presented in an intelligible fashion and written in standard English?

Reviewer #1: Yes

6. Review Comments to the Author

Reviewer #1: (No Response)

7. PLOS authors have the option to publish the peer review history of their article (what does this mean?). If published, this will include your full peer review and any attached files.

Reviewer #1: No

---

## [Editor Report · Acceptance letter]

27 Jun 2022

PONE-D-21-28346R2 

Multi-Class Classification Algorithms for the Diagnosis of Anemia in an Outpatient Clinical Setting 

Dear Dr. Vohra:

I'm pleased to inform you that your manuscript has been deemed suitable for publication in PLOS ONE. Congratulations! Your manuscript is now with our production department. 

Kind regards, 

on behalf of

Dr. Wajid Mumtaz 

Academic Editor

PLOS ONE